# The Impact of Perceived Sleep, Mood and Alcohol Use on Verbal, Physical and Sexual Assault Experiences among Student Athletes and Student Non-Athletes

**DOI:** 10.3390/ijerph18062883

**Published:** 2021-03-11

**Authors:** Jonathan Charest, Celyne H. Bastien, Jason G. Ellis, William D. S. Killgore, Michael A. Grandner

**Affiliations:** 1School of Psychology, Laval University, Québec City, QC G1V 0A6, Canada; Celyne.Bastien@psy.ulaval.ca; 2Centre for Sleep & Human Performance, Calgary, AL T2X 3V4, Canada; 3Department of Psychology, Northumbria University Newcastle, Sutherland Building, Newcastle-Upon-Tyne NE1 8ST, UK; jason.ellis@northumbria.ac.uk; 4Department of Psychiatry, University of Arizona, Tucson, AZ 85721, USA; killgore@email.arizona.edu (W.D.S.K.); grandner@email.arizona.edu (M.A.G.)

**Keywords:** sleep, student athletes, mental health

## Abstract

Previous research has shown that student athletes are more likely to be involved in a physical altercation or be a victim of verbal, physical and/or sexual abuse than student non-athletes, which can have long-lasting negative effects on mood, behavior and quality of life. In addition, among college students, sleep difficulties are ubiquitous and may deteriorate the unique life experience that university represents. The influences of poor sleep quality, mood and alcohol consumption related to these events are examined here between student athletes and student non-athletes. A series of hierarchical logistic regressions explored the relationship between verbal, physical and sexual assault risk factors. Results suggest that poor sleep, alcohol consumption and mood are all associated with exposure to a physical altercation or episode of abuse, irrespective of athlete status. Results also show that variables targeting self-reported difficulty sleeping and experiences of verbal, physical and sexual assault were positively associated. However, given the cross-sectional nature of the study, it is impossible to establish the direction of these relationships.

## 1. Introduction

College students are at risk for verbal, physical and sexual assault during their time at school, and these assaults can have long-lasting negative effects on mood, behavior and quality of life [1,2]. In American colleges, approximately 696,000 students are victims of physical assaults each year, and an additional 97,000 are victims of sexual assaults [2]. While the problem is believed to be pervasive and systemic in the culture of post-secondary college life, Frintner and Rubinson first described the prevalence to be higher for student athletes as opposed to non-athlete students [3]. This would suggest that the problem has been present for many years and remains unresolved. Furthermore, the Association of American Universities (AAU) released in 2015 results of their survey on sexual assault. The report showed that 11.7% of students reported non-consensual sexual contact, and 15% of perpetrators were student athletes [4]. Another core feature of collegiate students and student athletes is a tendency toward binge drinking [5,6,7,8]. The literature has been unequivocal on that matter; collegiate student athletes are more likely to binge drink than their student non-athlete counterparts [5,6,7,8]. Furthermore, according to the United Educators, 78% of sexual assaults were preceded by large alcohol consumption [4]. In addition to binge drinking, athletic culture may also play a major role in the over-representation of student athletes in sexual assaults [4]. For example, in 25% of multiple perpetrating assaults by athletes, the institution never investigated if the perpetrator was part of a larger team culture [4]. In addition, McCray’s narrative review of the literature on intercollegiate athletes and sexual violence also suggested an overrepresentation of student athletes being perpetrators in sexual assault cases [9]. It has been reported that 35% of the perpetrators are student athletes, while they only account for 3% of the entire student population [9].

The proportion of college students consuming alcohol is similar to the proportion reporting low sleep quality [10]. In fact, research identified a significant association between alcohol misuse among college students and poor sleep [10,11]. College students, as a group, tend to report irregular and insufficient sleep [12,13], with 12% to 14% reporting clinically significant symptoms of sleep disorder [12] and as many as 60% reporting poor sleep quality [11,14]. Moreover, research has continuously shown that student athletes are more likely to binge drink alcohol than the general student population [5,6,15]. The National Institute on Alcohol Abuse and Alcoholism defines binge drinking as a pattern of drinking alcohol that brings blood alcohol concentration (BAC) to 0.08%—or 0.08 g of alcohol per deciliter—or higher. For a typical adult, this pattern corresponds to consuming 5 or more drinks (male), or 4 or more drinks (female), in about 120 min [16]. Recently, Bastien and colleagues also showed that student athletes significantly engage in more binge drinking than their peer student non-athletes [17]. This is particularly concerning, as impaired sleep quality may increase the consequences of alcohol consumption among college students [16]. Sleep deprivation tends to decrease inhibition [18], impair decision making [19,20] and can even affect moral judgment [21]. However, the direction of the relationship remains unclear.

Sleep and depression are also interrelated—disturbed sleep is a cardinal feature of depression [22]. Approximately 14.8% of students report a diagnosis of depression during college years, and an estimated 11% will experience suicidal ideation [23]. Greater depressive symptoms have been linked to an irregular sleep schedule, while prolonged sleep latency was associated with loss of pleasure, self-punishment feelings and self-dislike [24]. Again, the directional nature of these associations remains indeterminate.

While sleep problems are ubiquitous among the average college student population, student athletes face even greater challenges regarding the impact of poor sleep habits on their mental health and performance [25]. The aim of the present study was to explore the possible association between subjective sleep difficulties, alcohol intake and mood on the likelihood of being a victim of verbal, physical and/or sexual abuse in a sample of college/university students. Because poor sleep, alcohol consumption and depressed mood are each independently associated with an increased disinhibition, we expect that these behaviors should be additive when combined, leading to an increased likelihood of a history of verbal, physical and sexual abuse. In addition, because student athletes report even greater sleep difficulties [26], we expect that these associations might be stronger in athletes versus student non-athletes. Therefore, because sleep difficulties and alcohol consumption have been reported to be greater in student athletes, we posit that the association between sleep complaints and verbal, physical and sexual assaults experienced within the preceding year would be stronger among student athletes than among the general student population.

## 2. Materials and Methods

### 2.1. Data Source

Data from the National College Health Assessment (NCHA) were used. The NCHA is an annual survey conducted by the American College Health Association (ACHA) to document prevalence and changes in a wide range of health-related factors among college students. Complete information about this dataset is available online (https://www.acha.org/NCHA) (accessed on 20 November 2020). This survey provides the largest known data source on health factors among American college and university students. Surveys were administered on paper or online. As part of the sampling methods, no student was entered into the sample twice, even if their university participated in more than one year’s survey. Additionally, institutions are kept anonymous, as are the individuals, in order to promote honest reporting. Survey data from 2011 to 2014 were used, as items did not change during this time period. Data were obtained from 44, 51, 57 and 34 colleges/universities in 2011, 2012, 2013 and 2014, respectively. This resulted in data from N = 27,774 in 2011, N = 28,237 in 2012, N = 32,964 in 2013 and N = 25,841 in 2014, a total of 111,498. Because retrospective data are under scrutiny in the present study and publicly available, no ethical approval was necessary nor was obtained.

### 2.2. Measures

Identification of varsity athlete status is self-reported (non-athlete or athlete). This status was assessed with the item, “Do you participate in organized college athletics?”. Responses were “Varsity”, “Club Sports” or “Intramurals”.

Perceived sleep difficulties were assessed with the item, “Within the last 12 months, have any of the following been traumatic or very difficult for you to handle?” with “Sleep Difficulties” being one of the items asked. Responses were “Yes” or “No”. 

Participants were also asked, “Within the last 12 months, did you experience any of the following”: “Were you in a physical fight?” (fight), “Were you physically assaulted (do not include sexual assault)?” (physical assault), “Were you verbally threatened?” (threat), “Were you sexually touched without your consent?” (touch), “Was sexual penetration attempted (vaginal, anal, oral) without your consent?” (sexual assault), “Were you sexually penetrated (vaginal, anal, oral) without your consent?” (rape), and “Were you a victim of stalking (e.g., waiting for you outside your classroom, residence, or office; repeated emails/phone calls)?” (stalking). All responses were “Yes” or “No”.

Depressed mood was assessed with the item, “Felt very sad” in the last 30 days. This was either “Yes” or “No”. Alcohol use was assessed with the item, “The last time you partied / socialized, how many drinks of alcohol did you have? (If you did not drink alcohol, please enter 0).” Responses were recorded in whole numbers and categorized as 0 (No drinks), 1–6 drinks, 7–16 drinks or 17 or more drinks. Additional covariates included age, sex and survey year.

### 2.3. Statistical Analyses

All variables were assessed using descriptive statistics (mean and standard deviation for continuous variables and percentages for categorical variables). To determine whether sleep disturbance was associated with assault variables (fight, physical assault, threat, touch, sexual assault, rape and stalking), hierarchical logistic regression analyses were used, with sleep difficulty as an independent variable and assault outcome as a dependent variable. Models included (1) unadjusted, (2) adjusted for age, sex and survey year and (3) adjusted by age, sex, survey year, depression and alcohol use. To evaluate whether the relationship between sleep and outcomes depended on depressed mood, sleep–depression interactions were evaluated. Significant sleep–depression interactions (*p* < 0.01) were seen for fight, physical assault, threat, touch, sexual assault and stalking. To evaluate whether the relationship between sleep and outcomes depended on alcohol intake, sleep–alcohol interactions were evaluated. Significant sleep–alcohol interactions (*p* < 0.01) were seen for physical assault, threat, touch, rape and stalking. Although there were no significant sleep–athlete interactions on outcomes (except for the case of physical assault), we chose to report results stratified by athlete status alongside combined results. The combined results reflect the findings that in general, relationships did not statistically differ by athlete status. However, given the unbalanced nature of the sample (only about 8% being athletes) and the heuristic value of being able to specifically describe these two groups separately, we have included results stratified by athlete status. All analyses were performed using STATA 14.0 (STATA Corp., College Station, TX, USA).

## 3. Results

### 3.1. Characteristics of the Sample

Characteristics of the sample are presented in Table 1. The sample included N = 111,498 students, including 7.79% (N = 8683) athletes. The sample was 66.63% (N = 74,291) female, depressed mood was present in 38.55% (N = 42,982), alcohol use was present in 62.39% (N = 69,563) and sleep difficulty was present in 25.68% (N = 28,632). Regarding verbal, physical and sexual assault variables, the most reported were verbal threats 17.83% (N = 19,880), and the least common was rape 1.65% (N = 1839). When athletes were compared to student non-athletes, all variables were significantly different between groups, though differences were clinically small. 

### 3.2. Verbal, Physical and Sexual Assaults: Their Association with Sleep Difficulties and Difference between Students Non-Athletes and Student Athletes

Relationships between perceived sleep difficulties and assault variables are reported in Table 2. In unadjusted analyses, having sleep difficulties was associated with an increased likelihood of all assault variables. In addition, after adjustment for age, sex, survey year, depressed mood and alcohol, being a student athlete and having sleep difficulties compared to student non-athletes was associated with a similar likelihood of all assault variables.

### 3.3. Association between Depressed Mood and Sleep Difficulties: Three Categories of Student Status: Combined, Student Non-Athlete and Student Athlete (Controlling for Alcohol Use)

Perceived sleep difficulties–depressed mood interaction was statistically significant, so results were stratified by depressed mood. Results of stratified analyses are reported in Table 3. In unadjusted analyses, those with no depressed mood were associated with an increased likelihood of all assaults compared to those with depressed mood. However, after adjustment for age, sex, survey year and alcohol, combined students with or without depressed mood were associated with an increased likelihood of all assaults compared to those without perceived sleep difficulties. However, the only category that was not associated with an increased likelihood was student athletes with a depressed mood for the outcome of rape. All other stratifications of combined students, student non-athletes and student athletes with or without depressed mood were associated with an increased likelihood of being involved in an assault compared to those without perceived sleep difficulties.

### 3.4. Association between Increasing Alcohol Use and Sleep Difficulties: Three Categories of Student Status: Combined, Student Non-Athlete and Student Athlete (Controlling for Depressed Mood)

Perceived sleep difficulties–alcohol interaction was also significant, so results were stratified by no alcohol use, 1–6 drinks, 7–16 drinks and 17 or more drinks per episode. These results are reported in Table 4. In unadjusted analyses, every alcohol use stratification was associated with an increased likelihood of all assaults. 

However, after adjustment for age, sex, survey year and depressed mood, being a student non-athlete was associated with an increased likelihood of every assault category across every alcohol stratification except for rape after a consumption of 17 drinks or more compared to those without perceived sleep difficulties. In addition, being a student athlete who did not drink was associated with an increased likelihood of every outcome’s category except for rape compared to those without perceived sleep difficulties. Moreover, being a student athlete that consumed between 1 and 6 drinks was associated with an increased likelihood of being in a fight, physical assault, threat, unwanted touching, sexual assault and stalking compared to those without perceived sleep difficulties. Being a student athlete that consumed between 7 and 16 drinks was associated with an increased likelihood of every assault except for rape. Lastly, being a student athlete who consumed 17 or more drinks was associated with an increased likelihood of being in a fight and physical assault compared to those without perceived sleep difficulties. 

## 4. Discussion

In this study, we evaluated whether student athletes were more likely to experience verbal, physical and/or sexual assault and to what degree these are related to perceived sleep difficulties and depend on depressed mood and alcohol use. Overall, few differences were found between student athletes and student non-athletes with perceived sleep difficulties on assault outcomes. The likelihood of assault outcomes may vary between student athletes and student non-athletes but does not differ statistically as shown in Table 2. Therefore, these results should be interpreted with caution even if, from a clinical perspective, an increased likelihood of 90% would warrant further consideration and attention. Moreover, consistent with a recent study, student athletes were found to consume large quantities of alcohol [17]. However, our results indicated that student athletes with perceived sleep difficulties that did not consume alcohol had an increased likelihood to experience 6 of the 7 assault outcomes investigated compared to 7 out of 7 for student non-athletes. Even if differences were not statistically significant, student athletes had a higher odds ratio, allegedly indicating that they were more likely to experience physical assault (71%) and unwanted touching (61%) and to have been stalked (34%), whereas student non-athletes were more likely to experience fighting, threats and sexual assault. Non-athlete students were more likely to experience forced sexual intercourse (rape) if they did not drink, whereas being a student athlete was not associated with an increased likelihood of experiencing forced sexual intercourse (rape). In addition, student athletes with perceived sleep difficulties who consumed 1 to 6 drinks were more likely to have reported an involvement in 5 of the 7 assault outcomes investigated compared to 7 out of 7 for non-athlete students. Interestingly, our results showed a linear increased likelihood of being involved in a fight among student athletes with perceived sleep difficulties when the number of alcoholic beverages increased (1-fold to 3-fold), while among student non-athletes, a slight linear decrease was observed (88% to 68%). These results may indicate a different impact of alcohol intake on student athletes and student non-athletes. Excessive drinking, which refers to 17 or more drinks, seems to exacerbate the aggressiveness of student athletes, as shown by the linear increase in the fighting per drinking category. However, it should be noted that none of these categories reached a statistical difference between student athletes and student non-athletes and should therefore be interpreted with caution. On the other hand, regardless of the number of alcoholic beverages, student athletes with perceived sleep difficulties did not differ from those without perceived sleep difficulties for forced sexual intercourse (rape). However, being a non-athlete student with perceived sleep difficulties is associated with an increased likelihood of forced sexual intercourse (rape). This was true for all categories of drinks apart from consuming 17 or more drinks.

Our results also indicated that student athletes and student non-athletes, reporting not being sad 30 days before the survey, were more likely to experience all assaults compared to those who did not have perceived sleep difficulties. Even if differences were not statistically significant, combined students, student athletes and student non-athletes with no depressed mood all reported a higher odds ratio in every assault category compared to those with a depressed mood. Student athletes with perceived sleep difficulties and a depressed mood were more likely to be involved in 6 out of 7 assaults except for forced sexual intercourse (rape), whereas student non-athletes were more likely to experience every assault compared to those without perceived sleep difficulties. Interestingly, all these relationships were stronger among those without a depressed mood for both student athletes (15–103%) and student non-athletes (22–43%). Even if student athletes and student non-athletes did not differ statistically in almost every category, our results demonstrated that the feeling of sadness may have impacted the two populations differently. For example, student athletes who did not feel sad were more likely (103%) to be involved in forced sexual intercourse (rape) compared to those feeling sad. For student non-athletes, there was no statistical difference between feeling sad or not (16%) for the forced sexual intercourse (rape) outcome. Surprisingly, our results are inconsistent with prior work suggesting that student athletes represent a subgroup that is more susceptible to engaging in dating violence [27,28]. It is estimated that 19% of male student athletes account for all sexual violence cases reported to Judicial Affairs offices [29]. However, it is important to note that research on sexual violent behavior is limited and is highly controversial, with studies drawing different conclusions [27]. Therefore, our results should be cautiously interpreted given that additional thorough research will be needed to establish with more precision the risk factors differentiating student athletes and student non-athletes.

These results illustrate that the life experiences of student athletes often differ from those of the non-athlete student populations. Athletes are a subpopulation among college students that is more likely to face sleep deprivation challenges due to training schedules, available training times, long trips to competitions, jet lag and pre-event anxiety [26,30]. However, our results surprisingly showed that sleep difficulties were more prevalent among student non-athletes compared to student athletes. The question regarding sleep unfortunately did not represent the severity of sleep difficulties, and it is unknown whether athletes were surveyed during their athletic season or while off-season, which could have a significant impact on their sleep perception. The potential role of sleep difficulties appears to be a significant contributor to the different type of assaults among student athletes and student non-athletes and should be further explored, particularly as it combines with excessive alcohol use [10,11]. Future studies could explore the potential usefulness of a well-structured sleep program administered early in the college curriculum for improving sleep among college students.

We also found that the presence of perceived sleep difficulties had a significant impact on all physical and sexual assault variables, across all alcohol conditions. Surprisingly, unhealthy and potential harmful situations seem to be impacted differently by the number of alcoholic beverages consumed by student athletes compared to student non-athletes. As previously mentioned, the number of beverages seems to impact the likelihood of being involved in forced sexual intercourse (rape) for student non-athletes but not for student athletes, although it could be argued that student athletes experience higher levels of sleep deprivation [30,31,32], which could potentially translate into a range of negative outcomes and that alcohol could exacerbate certain behaviors. However, it seems that the student non-athlete population may be equally impacted by the perceived sleep difficulties as shown by our results. Our results are inconsistent with prior work suggesting that insufficient sleep increases the risk of initiating reprehensible behavior, especially among heavier drinkers [4]. In fact, in our sample of student athletes, those who reported that they did not drink experienced more assaults than those who reported heavier drinking. Interestingly, threat, unwanted touching and sexual assault were more common among non-drinkers compared to heavy drinkers. It was previously demonstrated that the severity of sleep deprivation was the main contributor toward reprehensible behavior including fighting, sexual risk taking, smoking and binge drinking [32]. Therefore, our findings may suggest that perceived sleep difficulties could have been the central contributor of the aforementioned behaviors, regardless of alcohol consumption. Furthermore, there may be additional variables, such as personality traits [33] or past history [22], that were not accounted for which may be contributing to the reprehensible behavior among those who abstained from alcohol.

Among student non-athletes and student athletes, sleep disturbances are clinically relevant for the treatment and evaluation of mental health [34]. It is well established that sleep difficulties and depressed mood are intertwined [24]. In addition, difficult life events amongst student non-athletes and student athletes, such as competitive and academic failure, might result in negative emotions such as depressed mood [35], which could lead to adaptative behaviors including social avoidance [36,37,38,39]. It can be argued that the state of depression would inhibit any form of conflictual attitude or energized confrontational behavior due to this dysfunctional social adaption. Moreover, it has also been shown that sleep difficulties were closely linked to anger and aggression [34]. Regardless of the type of personality, Krizan and colleagues suggested that sleep loss equally influences the level of anger [34]. As a result of sleep loss, individuals may intensify aggressive responses [34]. Therefore, it could be assumed that perceived sleep difficulties combined with a depressed mood may have triggered social avoidance, while on the other hand, the absence of a depressed mood may have resulted in the failure of self-regulation that may subsequently contribute to aggression and reprehensible behavior. However, given our large simple, significant differences were anticipated between individuals with and without perceived sleep difficulties. Regardless of the lack of statistically significant differences between student athletes and student non-athletes, the smallest clinical improvement regarding mental health must be considered seriously, and with the literature on student athletes remaining sparse, the collegiate population would greatly benefit from a thorough investigation to develop a strategic and individualized approach for both student athletes and student non-athletes.

### Limitations

There are several limitations to the present study. First, the single sleep item included in the questionnaire was not taken from a validated sleep instrument. Thus, its reliability and validity have not been determined. With that in mind, results should be interpreted with appropriate caution. Second, the cross-sectional nature of the study precludes any inferences of causality. It may be the case that poor sleep leads to a greater propensity to encounter these kinds of situations (verbal, physical and sexual assault), or it may be the case that being exposed to these types of harmful or traumatic experiences may cause sleep disturbances. Alternatively, sleep loss may lead to disinhibition, which itself may lead to poor sleep (directly and indirectly through increased drinking, depressed mood or the consequences of poor decisions). Third, these data were all provided through self-report. Therefore, there is no objective verification of the responses. Fourth, the competitive level of the athletes was not reported. Hence, it is unknown whether athletes were Division I, II or III, which may play a role in athletic, academic and/or social factors related to the outcomes of this study. Moreover, the only institution level variable is school type (public/private), which limits the possible assessment of the random effects per institution. It should be considered in future research to have a school code entered to provide hypothetical differences between institutions to a deeper level than public and private institutions. Moreover, future research should handle alcohol intake as a continuous variable for a better understanding of its impact on assaults and should use the same time range for sleep and depression items. In addition, consideration should be given to dividing the type of alcohol into three categories (beer, wine and spirits). These categories could be beneficial to better understand the relationship between students, student athletes and alcohol consumption. Another limitation is the dichotomizing of depressed mood based on the question “I feel very sad”. Although this is a common item used in many psychometric scales which assess depression—e.g., the Beck Depression Inventory—sad affect is only one component of depression, and future research would benefit from a more holistic measure of depression which accounts for broader range of symptom domains. Additionally, while the data do show the links between poor sleep, alcohol and assaults, many other factors, related to stress and/or trauma, may have influenced these findings. Future research should examine a broad framework of stressors to determine their impact on these relationships.

## 5. Conclusions

The present study found that self-reported difficulty sleeping and experiences of verbal, physical and sexual assault were positively associated. However, given the cross-sectional nature of the study, it is impossible to establish the direction of these relationships. Nevertheless, results suggest that sleep disturbances may be an independent risk factor of being involved in a situation where an assault of any kind (verbal, physical or sexual) is more likely to happen. On the other hand, being involved in assaults (verbal, physical or sexual) could also trigger sleep difficulties. Future research should aim to determine whether the number of assaults can be reduced by improving sleep health and what the impacts of an assault are on sleep difficulties. Ultimately, this could also pave the way to bidirectional research on the impact of these assaults on sleep difficulties and the potential role of post-traumatic stress disorder (PTSD) following an assault. In addition, it would be fundamental to investigate the different risk factors that can contribute to these different assaults, which could lead to the development of preventive and educational interventions. Ultimately, interventions aimed at educating students about the importance of sleep and the potentially harmful effects of sleep loss combined with alcohol use seem warranted.

## Figures and Tables

**Table 1 ijerph-18-02883-t001:** Characteristics of the sample.

Variable	Category/Units	Complete Sample	Non-Athlete	Athlete	*p*
N		111,498	102,815	8683	<0.0001
Age	Age	21.5 (3.6)	21.7 (3.7)	19.6 (1.9)	<0.0001
Sex	Male	33.37%	32.93%	38.51%	<0.0001
	Female	66.63%	67.07%	61.49%	<0.0001
Year	2011	24.33%	24.20%	25.94%	<0.0001
	2012	24.42%	24.23%	26.70%	<0.0001
	2013	28.53%	28.49%	29.01%	<0.0001
	2014	22.71%	23.08%	18.36%	<0.0001
Depressed mood	Not in last 30 days	61.45%	61.05%	66.10%	<0.0001
	Yes in last 30 days	38.55%	38.95%	33.90%	<0.0001
Alcohol	None	37.61%	37.63%	37.34%	<0.0001
	1–6	33.30%	33.10%	35.68%	<0.0001
	6–16	25.44%	25.53%	24.40%	<0.0001
	17 or More	3.65%	3.74%	2.58%	<0.0001
Sleep difficulties	Yes	25.68%	26.17%	19.81%	<0.0001
Fight	Yes	5.29%	5.03%	8.36%	<0.0001
Physical assault	Yes	3.49%	3.43%	4.20%	0.0002
Threat	Yes	17.83%	17.57%	20.98%	<0.0001
Touch	Yes	6.36%	6.28%	7.24%	0.0005
Sexual assault	Yes	2.75%	2.68%	3.53%	<0.0001
Rape	Yes	1.65%	1.62%	2.11%	0.0005
Stalk	Yes	5.54%	5.49%	6.03%	0.0386

**Table 2 ijerph-18-02883-t002:** Verbal, physical and sexual assaults: their association with sleep difficulties and differences between students and student athletes.

OUTCOMES	Unadjusted			Adjusted ^1^			Fully Adjusted ^2^		
	OR	95% CI	*p*	OR	95% CI	*p*	OR	95% CI	*p*
**COMBINED STUDENTS**									
Fight	1.815	(1.720, 1.916)	<0.0001	2.008	(1.899, 2.124)	<0.0001	1.810	(1.706, 1.921)	<0.0001
Physical Assault	2.366	(2.219, 2.523)	<0.0001	2.406	(2.253, 2.570)	<0.0001	2.060	(1.923, 2.208)	<0.0001
Threat	2.200	(2.130, 2.272)	<0.0001	2.326	(2.250, 2.405)	<0.0001	2.029	(1.959, 2.101)	<0.0001
Touching	2.256	(2.148, 2.369)	<0.0001	2.181	(2.074, 2.292)	<0.0001	1.838	(1.743, 1.937)	<0.0001
Sexual Assault	2.443	(2.273, 2.625)	<0.0001	2.334	(2.168, 2.513)	<0.0001	1.960	(1.814, 2.118)	<0.0001
Rape	2.657	(2.423, 2.912)	<0.0001	2.520	(2.293, 2.769)	<0.0001	2.095	(1.899, 2.313)	<0.0001
Stalking	2.519	(2.392, 2.654)	<0.0001	2.446	(2.320, 2.579)	<0.0001	2.195	(2.077, 2.320)	<0.0001
**STUDENT NON-ATHLETES**									
Fight	1.842	(1.739, 1.953)	<0.0001	2.015	(1.898, 2.139)	<0.0001	1.809	(1.698, 1.927)	<0.0001
Physical Assault	2.309	(2.156, 2.472)	<0.0001	2.346	(2.188, 2.515)	<0.0001	1.997	(1.857, 2.149)	<0.0001
Threat	2.213	(2.139, 2.289)	<0.0001	2.330	(2.250, 2.412)	<0.0001	2.032	(1.959, 2.108)	<0.0001
Touching	2.250	(2.137, 2.369)	<0.0001	2.177	(2.066, 2.295)	<0.0001	1.836	(1.737, 1.940)	<0.0001
Sexual Assault	2.479	(2.296, 2.676)	<0.0001	2.363	(2.186, 2.555)	<0.0001	1.976	(1.821, 2.145)	<0.0001
Rape	2.695	(2.444, 2.972)	<0.0001	2.560	(2.317, 2.829)	<0.0001	2.138	(1.926, 2.374)	<0.0001
Stalking	2.505	(2.372, 2.645)	<0.0001	2.432	(2.301, 2.571)	<0.0001	2.189	(2.066, 2.320)	<0.0001
**STUDENT ATHLETES**									
Fight	1.794	(1.513, 2.127)	<0.0001	2.035	(1.699, 2.438)	<0.0001	1.874	(1.547, 2.271)	<0.0001
Physical Assault	2.991	(2.404, 3.722)	<0.0001	3.061	(2.443, 3.836)	<0.0001	2.776	(2.189, 3.520)	<0.0001
Threat	2.172	(1.929, 2.445)	<0.0001	2.374	(2.099, 2.686)	<0.0001	2.051	(1.801, 2.336)	<0.0001
Touching	2.338	(1.962, 2.786)	<0.0001	2.192	(1.831, 2.623)	<0.0001	1.853	(1.534, 2.239)	<0.0001
Sexual Assault	2.165	(1.692, 2.770)	<0.0001	1.966	(1.523, 2.547)	<0.0001	1.698	(1.298, 2.220)	<0.0001
Rape	2.207	(1.610, 3.026)	<0.0001	1.967	(1.414, 2.736)	<0.0001	1.609	1.141, 2.270)	<0.0001
Stalking	2.932	(2.432, 3.533)	<0.0001	2.764	(2.283, 3.347)	<0.0001	2.401	1.966, 2.933)	<0.0001

^1^ Adjusted for age, sex and survey year; ^2^ Adjusted for age, sex, survey year, depressed mood and alcohol use.

**Table 3 ijerph-18-02883-t003:** Association between depressed mood and sleep difficulties: three categories of student status: combined, student non-athlete and student athlete (controlling for alcohol use).

OUTCOMES	No Depressed Mood			Yes Depressed Mood		
	OR	95% CI	*p*	OR	95% CI	*p*
**COMBINED STUDENTS**						
Fight	1.963	(1.803, 2.137)	<0.0001	1.678	(1.546, 1.820)	<0.0001
Physical Assault	2.346	(2.112, 2.605)	<0.0001	1.847	(1.712, 2.051)	<0.0001
Threat	2.221	(2.108, 2.339)	<0.0001	1.884	(1.798, 1.973)	<0.0001
Touching	2.021	(1.857, 2.200)	<0.0001	1.734	(1.623, 1.853)	<0.0001
Sexual Assault	2.152	(1.894, 2.445)	<0.0001	1.862	(1.691, 2.050)	<0.0001
Rape	2.292	(1.946, 2.699)	<0.0001	1.997	(1.767, 2.257)	<0.0001
Stalking	2.400	(2.209, 2.607)	<0.0001	2.047	(1.903, 2.202)	<0.0001
**STUDENT NON-ATHLETES**						
Fight	1.988	(1.814, 2.178)	<0.0001	1.660	(1.523, 1.810)	<0.0001
Physical Assault	2.267	(2.026, 2.536)	<0.0001	1.828	(1.663, 2.009)	<0.0001
Threat	2.214	(2.096, 2.338)	<0.0001	1.895	(1.805, 1.990)	<0.0001
Touching	1.979	(1.809, 2.165)	<0.0001	1.754	(1.636, 1.881)	<0.0001
Sexual Assault	2.136	(1.864, 2.448)	<0.0001	1.893	(1.711, 2.096)	<0.0001
Rape	2.247	(1.887, 2.675)	<0.0001	2.083	(1.830, 2.372)	<0.0001
Stalking	2.385	(2.186, 2.602)	<0.0001	2.047	(1.896, 2.210)	<0.0001
**STUDENT ATHLETES**						
Fight	1.952	(1.507, 2.527)	<0.0001	1.797	(1.348, 2.394)	<0.0001
Physical Assault	3.171	(2.290, 4.392)	<0.0001	2.417	(1.722, 3.393)	<0.0001
Threat	2.378	(1.972, 2.868)	<0.0001	1.792	(1.501, 2.141)	<0.0001
Touching	2.482	(1.871, 3.292)	<0.0001	1.498	(1.171, 1.916)	0.001
Sexual Assault	2.090	(1.390, 3.142)	<0.0001	1.477	(1.044, 2.090)	0.028
Rape	2.324	(1.370, 3.940)	0.002	1.286	(0.832, 1.988)	0.257
Stalking	2.760	(2.059, 3.702)	<0.0001	2.120	(1.623, 2.770)	<0.0001

Adjusted for age, sex, survey year and alcohol use.

**Table 4 ijerph-18-02883-t004:** Association between increasing alcohol use and sleep difficulties: three categories of student status: combined, student non-athlete and student athlete (controlling for depressed mood).

OUTCOMES	COMBINED STUDENTS			STUDENT NON-ATHLETES			STUDENT ATHLETES		
	OR	95% CI	*p*	OR	95% CI	*p*	OR	95% CI	*p*
**NO DRINKS**									
Fight	1.869	(1.660, 2.104)	<0.0001	1.887	(1.663, 2.142)	<0.0001	1.606	(1.086, 2.375)	0.018
Physical Assault	2.224	(1.940, 2.549)	<0.0001	2.163	(1.873, 2.499)	<0.0001	2.877	(1.802, 4.594)	<0.0001
Threat	2.227	(2.091, 2.372)	<0.0001	2.234	(2.092, 2.387)	<0.0001	2.156	(1.720, 2.732)	<0.0001
Touching	2.058	(1.841, 2.300)	<0.0001	1.992	(1.771, 2.240)	<0.0001	2.603	(1.779, 3.807)	<0.0001
Sexual Assault	2.145	(1.811, 2.541)	<0.0001	2.155	(1.798, 2.582)	<0.0001	1.991	(1.145, 3.462)	0.015
Rape	1.953	(1.584, 2.407)	<0.0001	2.005	(1.606, 2.503)	<0.0001	1.619	(0.814, 3.219)	0.169
Stalking	2.204	(2.001, 2.429)	<0.0001	2.179	(1.969, 2.413)	<0.0001	2.523	(1.764, 3.611)	<0.0001
**1-6 DRINKS**									
Fight	1.764	(1.582, 1.968)	<0.0001	1.786	(1.591, 2.006)	<0.0001	1.775	(1.249, 2.523)	0.001
Physical Assault	2.120	(1.875, 2.397)	<0.0001	2.100	(1.845, 2.390)	<0.0001	2.731	(1.797, 4.150)	<0.0001
Threat	1.974	(1.858, 2.096)	<0.0001	1.982	(1.861, 2.111)	<0.0001	2.066	(1.665, 2.565)	<0.0001
Touching	1.907	(1.746, 2.084)	<0.0001	1.939	(1.767, 2.128)	<0.0001	1.658	(1.206, 2.278)	<0.0001
Sexual Assault	2.057	(1.800, 2.351)	<0.0001	2.142	(1.860, 2.466)	<0.0001	1.448	(0.900, 2.330)	0.127
Rape	2.368	(1.995, 2.811)	<0.0001	2.500	(2.083, 2.999)	<0.0001	1.514	(0.832, 2.756)	0.175
Stalking	2.232	(2.030, 2.454)	<0.0001	2.247	(2.034, 2.483)	<0.0001	2.311	(1.663, 3.211)	<0.0001
**7-16 DRINKS**									
Fight	1.795	(1.626, 1.983)	<0.0001	1.786	(1.606, 1.986)	<0.0001	1.964	(1.425, 2.706)	<0.0001
Physical Assault	1.889	(1.678, 2.126)	<0.0001	1.793	(1.582, 2.033)	<0.0001	2.690	(1.789, 4.046)	<0.0001
Threat	1.893	(1.777, 2.017)	<0.0001	1.882	(1.762, 2.011	<0.0001	2.001	(1.576, 2.541)	<0.0001
Touching	1.644	(1.506, 1.794)	<0.0001	1.650	(1.506, 1.808)	<0.0001	1.556	(1.123, 2.157)	0.008
Sexual Assault	1.837	(1.624, 2.079)	<0.0001	1.815	(1.594, 2.067)	<0.0001	1.785	(1.157, 2.752)	0.009
Rape	2.023	(1.724, 2.374)	<0.0001	2.031	(1.716, 2.402)	<0.0001	1.630	(0.907, 2.927)	0.102
Stalking	2.095	(1.889, 2.325)	<0.0001	2.095	(1.879, 2.335)	<0.0001	2.204	(1.489, 3.260)	<0.0001
**17 OR MORE DRINKS**									
Fight	1.792	(1.472, 2.181)	<0.0001	1.680	(1.364, 2.070)	<0.0001	3.330	(1.579, 7.019)	0.002
Physical Assault	1.952	(1.534, 2.483)	<0.0001	1.865	(1.445, 2.407)	<0.0001	2.855	(1.154, 7.062)	0.023
Threat	1.990	(1.705, 2.322)	<0.0001	2.008	(1.711, 2.358)	<0.0001	1.527	(0.791, 2.951)	0.207
Touching	1.831	(1.461, 2.295)	<0.0001	1.801	(1.417, 2.288)	<0.0001	1.958	(0.878, 4.366)	0.100
Sexual Assault	1.579	(1.143, 2.181)	0.005	1.543	(1.094, 2.178)	0.014	1.082	(0.345, 3.398)	0.892
Rape	1.625	(1.092, 2.416)	0.016	1.500	(0.979, 2.300)	0.063	1.366	(0.351, 5.325)	0.653
Stalking	2.295	(1.796, 2.933)	<0.0001	2.241	(1.730, 2.903)	<0.0001	3.278	(1.300, 8.295)	0.012

Adjusted for age, sex, survey year and depressed mood.

## Data Availability

The data presented in this study are openly available at https://www.acha.org/NCHA (accessed on 15 December 2020).

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
