# Peer review of "The Impact of Perceived Sleep, Mood and Alcohol Use on Verbal, Physical and Sexual Assault Experiences among Student Athletes and Student Non-Athletes"

_ijerph, 2021, doi:10.3390/ijerph18062883_

Round 1

Reviewer 1 Report

The authors have addressed my remaining comments. Thank you for the opportunity to review this manuscript.

Reviewer 2 Report

The authors did a great job and the revised version is - in my opinion - now acceptable for publication.

This manuscript is a resubmission of an earlier submission. The following is a list of the peer review reports and author responses from that submission.

Round 1

Reviewer 1 Report

GENERAL COMMENTS:
Thank you for the opportunity to review this interesting paper. This manuscript focuses on verbal, physical and sexual assault experiences among student-athletes and non-athletes. Further, the study aimed at exploring the possible association between subjective sleep difficulties, alcohol intake and mood on the likelihood of a history of these experiences. In general, the research question is relevant as the knowledge on this topic is limited.

Overall, the manuscript is well written. However, I have some comments that are presented below for each section.

INTRODUCTION:

  • In general, Introduction is well written. The authors describe the prevalence of this kind of assault experiences in America. Does previous research provide information about the sex differences and differences between, for example, athletes in team and individual sports? This kind of information would be informative.

MATERIALS AND METHODS

  • Has the research appropriate ethical approvals and have written consent been obtained?
  • Page 2, line 85-86: Identification of athlete status belongs to the “Measures” section. I suppose this was one question in the questionnaire? If so, the description of this question with responses would be informative. As a non-American reader, I miss the definition what is meant by having athlete status.
  • Lack of validated measures is one major limitation of this study. However, the authors have discussed this adequately in the Discussion section. Especially, depressed mood and alcohol use were measured with very vague measures. Also sleep difficulties has been measured with a very vague question. In this context, it would have been important to measure the reasons for insufficient sleep. Is the reason that you go to bed too late (you spend your time with friends or with studying late in the evening/night) or is the reason that you go to bed early enough but you have insomnia.
  • It would be interesting to know in which context and by whom these assaults have been occurred but I guess the questionnaire did not include this kind of questions?
  • It seems that statistical analyses used are appropriate and sufficiently justified and explained.

RESULTS

  • Characteristics of the sample are presented in the text with “n” and percentage in parentheses. I would recommend to report this information other way round (for example page 3, line 123: …was present in 38,55% (n=42982)…)
  • Page 3, line 126-127: You could describe in which way the variable were different between the groups.
  • It would be very interesting to get some information about potential gender differences? Are there any differences between the genders, for example, just in the prevalence of experienced assaults?
  • Page 7, line 31: I guess there is a typo in the line 31 (on the other hand…).

DISCUSSION AND CONCLUSIONS

  • Page 8, line 51: compered to non-athletes?
  • Page 8, line 56-58: You describe that, based on previous studies, athletes are seen more likely to face sleep deprivations. In the present study, however, table 1 showed that sleep difficulties were more common among non-athletes. Should you discuss this, what do you think about this finding?
  • I agree with the authors, it would be important to have data on stressful or traumatic events that may have influenced the findings.
  • I don’t know if the data allows that, but it would have been interesting to know, if there are differences between different types of sports (for example team vs. individual sports). This could, however, be interesting question for future studies.
  • I’m not sure about your conclusions. You state that the sleep disturbances are a risk factor of being involved in a situation where an assault of any kind is more likely to happen. As a cross-sectional study, the findings do not provide evidence for causality. Why do you suggest that the sleep disturbances are a risk factor for assault experiences and not vice versa? Wouldn’t it be also logical to suggest that this kind of experiences could be reasons for sleep difficulties? In a same way, you state that the number of assaults can be reduced by improving sleep health. Wouldn’t it be more important to prevent this kind of assaults, to develop activities and strategies for college (and college sports) that no one has to experience verbal, physical and sexual assaults?

Author Response

Manuscript: IJERPH-1023751

Comments and Suggestions for Authors

Reviewer 1

We would like to thank reviewer 1 for the time and work devoted to revising this manuscript. We believe the comments have been helpful in enhancing our manuscript. We hope our responses and modifications will satisfy reviewer 1. In addition, if reviewer 1 has any additional comments and suggestions, we will be more than happy to add them to our manuscript.

INTRODUCTION

Comment 1: In general, Introduction is well written. The authors describe the prevalence of this kind of assault experiences in America. Does previous research provide information about the sex differences and differences between, for example, athletes in team and individual sports? This kind of information would be informative.

Response 1: Thank you for your comment. The difference between individual and team sports would be of high interest for the authors. However, we are not aware of such research. We believe this would be an important investigation that may shed light on the different culture across different sports.

MATERIALS AND METHODS

Comment 2: Has the research appropriate ethical approvals and have written consent been obtained?

Response 2:    Thank you for this comment. Given the retrospective nature of the data with did not had to obtained consent for this research. These data are publicly available at the provided address (https://www.acha.org/NCHA). The American College Health Association – National College Health Assessment (ACHA-NCHA) conduct survey yearly on different topic across the United States. Colleges that are interested to participate can contact the ACHA-NCHA research department to enroll their institution. The survey/research is then presented to the potential participant (students) and the consent is then presented. Therefore, we do not have access to the consent that was presented, and we used retrospective data.

Comment 3:   Page 2, line 85-86: Identification of athlete status belongs to the “Measures” section. I suppose this was one question in the questionnaire? If so, the description of this question with responses would be informative. As a non-American reader, I miss the definition what is meant by having athlete status.

Response 3:    We would like to thank the reviewer for this question. The reviewer is right when he states that this is a question and should therefore be moved to the measures section. The question is simply: Participated in organized college athletics? The possible choices are: Varsity; Club Sports; Intramurals. Students who do not engage in varsity or club sports at their colleges/universities can not have a status of student-athletes.

Comment 4:   Lack of validated measures is one major limitation of this study. However, the authors have discussed this adequately in the Discussion section. Especially, depressed mood and alcohol use were measured with very vague measures. Also sleep difficulties has been measured with a very vague question. In this context, it would have been important to measure the reasons for insufficient sleep. Is the reason that you go to bed too late (you spend your time with friends or with studying late in the evening/night) or is the reason that you go to bed early enough but you have insomnia.

Response 4:    We would like to thank the reviewer for this question. The lack of validated measures is the major limitation of this study and we believe this was addressed adequately. We do agree that future research project should explore the influential factors for insufficient or disturbed sleep. However, the retrospective nature of the data did not allow us to explore that question. This manuscript primary objective is to establish the prevalence of poor sleep in with a general, broad perspective, and we are hopeful it will pave the way for a subsequent project to investigate thoroughly the reasons of poor, inadequate and disturbed sleep with a distinction in regard to insomnia.

Comment 5:   It would be interesting to know in which context and by whom these assaults have been occurred but I guess the questionnaire did not include this kind of questions?

Response 5:    We agree with this reviewer that this information would have been of high interest. The reviewer is also right stating that this information was unfortunately unavailable simply because it was not included in the survey provided by the ACHA-NCHA. We hope that the results of this manuscript will instigate interest with other researcher to explore that question.

Comment 6:   It seems that statistical analyses used are appropriate and sufficiently justified and explained.

Response 6: Thank you for this remark. However, as you will notice, reviewer 2 made substantial recommendations on statistical analyses. Therefore, this section has received a lot of attention and a lot of modification have been made.

RESULTS

Comment 7:   Characteristics of the sample are presented in the text with “n” and percentage in parentheses. I would recommend to report this information other way round (for example page 3, line 123: …was present in 38,55% (n=42982)…)

Response 7:    We would like to thank the reviewer for this remark. We do agree that interchanging the “n” and “%” does facilitate the flow for the readership. We have modified in the text as suggested.

Comment 8:   Page 3, line 126-127: You could describe in which way the variable were different between the groups.

Response 8:    We are not sure what the reviewer is requesting or suggesting, we do apologize for this. We believe the reviewer is requesting a description between the student-athlete and non-athlete. These descriptions are included in Table 1 and the reason we have not wrote them is simply to not overcrowded this paragraph with several percentage and number that are available in Table 1 and easier to understand with a table than in sentences.

Comment 9: It would be very interesting to get some information about potential gender differences? Are there any differences between the genders, for example, just in the prevalence of experienced assaults?

Response 9:    Thank you for this comment. We agree with the reviewer that it would be interesting to add information about sex difference. The aim of the present manuscript was only to highlight the presence and establish the prevalence of assaults associated with sleep difficulties with different covariates (depressed mood, alcohol use and student status). We have noticed a significant difference between gender, but we believe that this will be a completely different paper. We believe and agree with this reviewer that it would be of high interest to establish which assault is more likely to be experience if you are a man or a woman. We believe that a follow up paper should address this in a different fashion so we can establish the direction of the relationship (different factor may impact sleep between men and women for example). With this paper we are hoping that the elevated odds ratio will interest other researcher to investigate this question further such as the aforementioned question of the difference between men and women in relationship with the different assaults.

Comment 10: Page 7, line 31: I guess there is a typo in the line 31 (on the other hand…).

Response 10: Thank you for this remark. We have replaced “author” by “the other hand”.

DISCUSSION AND CONCLUSIONS

Comment 11: Page 8, line 51: compered to non-athletes?

Response 11: We would like to thank the reviewer for this comment. We have added “compared to non-athletes” as requested. It will prevent the readership to go back and forth between the text and the tables.

Comment 12: Page 8, line 56-58: You describe that, based on previous studies, athletes are seen more likely to face sleep deprivations. In the present study, however, table 1 showed that sleep difficulties were more common among non-athletes. Should you discuss this, what do you think about this finding?

Response 12: We would like to thank the reviewer for this comment. We do agree with the reviewer that this finding should be further discussed. Rarely do we see non-athletes with more sleep difficulties compared to collegiate athlete. However, we believed that this is explained due to the vague question regarding sleep “Within the last 12 months, have any of the following been traumatic or very difficult for you to handle” ; Sleep difficulties? A lot of student-athletes do not consider experiencing sleep difficulties because they usually initiate sleep quickly since they are exhausted from their abnormal schedule. However, the ability to initiate sleep quickly does not necessarily represent good sleep. Therefore, with a validated questionnaire such as the Athlete Sleep Screening Questionnaire (ASSQ) or the Insomnia Severity Index (ISI) which asks specific question regarding sleep, we believed the results would have been different.

Comment 13: I agree with the authors, it would be important to have data on stressful or traumatic events that may have influenced the findings.

Response 13: We would like to thank the reviewer for this remark.

Comment 14: I don’t know if the data allows that, but it would have been interesting to know, if there are differences between different types of sports (for example team vs. individual sports). This could, however, be interesting question for future studies.

Response 14:  We go agree that this would have been an excellent variable to investigate. Unfortunately, the type of sports was not included in the survey provided by ACHA-NCHA. The difference between sports question is receiving more attention lately and we believe that in every future research investigating athlete and student-athlete, this variable /question should always be included.

Comment 15: I’m not sure about your conclusions. You state that the sleep disturbances are a risk factor of being involved in a situation where an assault of any kind is more likely to happen. As a cross-sectional study, the findings do not provide evidence for causality. Why do you suggest that the sleep disturbances are a risk factor for assault experiences and not vice versa? Wouldn’t it be also logical to suggest that this kind of experiences could be reasons for sleep difficulties? In a same way, you state that the number of assaults can be reduced by improving sleep health. Wouldn’t it be more important to prevent this kind of assaults, to develop activities and strategies for college (and college sports) that no one has to experience verbal, physical and sexual assaults?

Response 15: We would like to thank the reviewer for this comment. We apologize for the unfortunate confusion by our conclusion. We agree with the reviewer, the intent was not this report a causality but a relationship that could be bidirectional. We have reworded and restructured the conclusion to make it clear that we do not make any inference to a causality.

Reviewer 2 Report

IJERPH-1023751 “Verbal, physical, and sexual assault experiences among student-athletes and non-athletes: Impact of sleep, mood, and alcohol use”

In this manuscript, the authors report findings from a large survey conducted on health-related factors in college students. This is an important topic given the noted prevalence of abuse history, sleep problems, and alcohol use in college students (both athletes and non-athletes), as well as known impacts of these factors on later health and functioning. Study strengths include the large sample size across a number of different universities and assessment of multiple types of abuse.

However, I have major concerns regarding the description of study aims, analytic plan, and results, which prevent me from providing a recommendation about this paper’s fitness for publishing at the current time. Major concerns and additional suggestions for improvement are provided below.

Major Comments:

  1. The study aims do not appear to match the analytic plan. In Lines 66-68, the authors report their aim to “Explore the possible association between subjective sleep difficulties, alcohol intake and mood on the likelihood of a history of … abuse …” As currently written, I was expecting you to look at separate predictors of sleep, alcohol, and mood on abuse outcomes. In the analytic plan, it seems that alcohol use and depression are either covariates and/or moderators, and the authors also report examining “alcohol use and depressive mood as outcome variables.” If sleep difficulties are the primary predictor, please make this more clear throughout the document. Further, I am not sure what is meant by “additive when combined” (Line 70) – are you also comparing abuse history for students who met criteria for each of these variables vs. those who didn’t?

  1. Statistical Analyses section: There are several inconsistencies regarding with regard to the type of analyses used and description of the interaction terms. First, the Abstract notes that hierarchical logistic regressions were used, which would be appropriate given the clustering of students within schools, while Section 2.3 notes binomial logistic regressions. Second, in Lines 110-111 the authors note that they tested whether “athlete status interacts with sleep difficulties on assault…” and in line 114 they note “an interaction term for each assault variable by student-athlete status.” These statements suggest different things. Further, I am confused why alcohol use and depressive mood are listed as outcome variables. This is not how they are presented in the study aims, nor in the results. It is critical that this paragraph state exactly what how you conducted your analyses, what your predictor and outcome variables are, what variables are being tested as moderators, and what the covariates are for your analyses. At present, this is not clear.

  1. Results: The section headings for 3.3 and 3.4 are unclear. Did the authors test 3-way interactions with sleep X athlete status X depression (or alcohol)? That is what those titles, and the corresponding table titles indicate. Further, the interaction results are confusing. The authors note significant interactions by depressed mood, alcohol use, and athlete status, which is why they note that they stratified the results, yet Tables 3-5 and corresponding text present data that suggest the relationship between sleep and abuse was significant across depressed/non depressed groups, all alcohol categories, and both athletes and non-athletes. What should we make of this? What information did we learn from the stratified analyses? Did the authors test formal moderation, for example, using the guidelines put forth by Aiken and West in their seminal text on moderation? (Aiken, L. S., & West, S. G. (1991). Multiple regression: Testing and interpreting interactions. Newbury Park, CA: Sage). Additionally, throughout the Results section, I would refer to odds ratios that are greater than 2 by using “2-fold greater odds,” as results such as “102%” are difficult for readers to interpret. Finally, p values in the text are not necessary, as they are stated in the tables.

  1. The tables must be more clear regarding what is the predictor and what is the outcome and add a note with the covariates and the reference group. It would be helpful to including column headings such as “Abuse Outcomes” and “Sleep Difficulties”. Currently, the abuse variables as presented as if they are the predictors, and it is unclear what the “unadjusted, adjusted, fully-adjusted” columns refer to. Further, the titles are confusing. Tables 2-4 do not show results broken down by “student-athlete status.”

Other suggestions for improvement:

Abstract

  • Line 17 – what is meant by “deteriorate the climate”?

Introduction

  • Please specify the difference in prevalence of abuse/assault between athletes vs. non-athletes (e.g., from the AAU 2015 report or the Frintner article, etc.).
  • Lines 41-42: clarify if there is an overrepresentation of student-athletes as perpetrators or victims (or both?).
  • The Introduction describes data on perpetration of assaults, which is interesting but somewhat muddies the waters. I found myself not sure if you were studying victimization, perpetration, or both when I got to your hypotheses. I would consider revising the Introduction to be more clear and perhaps removing data on perpetration if it is not directly related to your hypotheses. This would also clarify what the authors mean by “increased likelihood of abuse” (line 71). Further, lines 75-76 refer to assaults within the past year – is that also the timeframe used in the first hypothesis?
  • Lines 69-70, 104: the phrase “poor cognition functioning” is odd, please clarify what this means and/or revise.

Method:

  • Why did you not adjust for race, given known differences in sleep and abuse by racial/ethnic background?
  • What was the rationale for categorizing alcohol use as stated?

Discussion

  • 8, Lines 41-45: I do not agree with the authors’ statements. Based on your results, the odds ratios for the relationship between sleep and abuse outcomes were almost identical (and all significant) across groups based on depression, alcohol, and athlete status. Thus, it is not accurate to say that those who did not drink or those who were not depressed showed a stronger relationship between sleep and certain categories of abuse. What are we to take away from these findings? They are directly in conflict with the rationale and aims of your study.
  • Line 97 – the authors refer to “sleep items,” but it appears that a single item was used in analysis. Along these lines, the use of single item should also be mentioned as a study limitation.
  • I appreciate the authors’ mention in Line 127 that although the differences between athletes and non-athletes were statistically significant (as would be expected from the large sample size), they were clinically small. Indeed, the raw percentages for most sample characteristics are quite similar for all variables. This needs to be expanded upon as a limitation and the statement in the Discussion (p. 8, Lines 37-40) needs to be tempered accordingly.

Author Response

Manuscript: IJERPH-1023751

Comments and Suggestions for Authors

Reviewer 2

We would like to thank reviewer 2 for the time and work devoted to revising this manuscript. We believe the comments have been helpful in enhancing our manuscript. We hope our responses and modifications will satisfy reviewer 2. In addition, if reviewer 2 has any additional comments and suggestions, we will be more than happy to add them to our manuscript.

Comment 1: The study aims do not appear to match the analytic plan. In Lines 66-68, the authors report their aim to “Explore the possible association between subjective sleep difficulties, alcohol intake and mood on the likelihood of a history of … abuse …” As currently written, I was expecting you to look at separate predictors of sleep, alcohol, and mood on abuse outcomes. In the analytic plan, it seems that alcohol use and depression are either covariates and/or moderators, and the authors also report examining “alcohol use and depressive mood as outcome variables.” If sleep difficulties are the primary predictor, please make this more clear throughout the document. Further, I am not sure what is meant by “additive when combined” (Line 70) – are you also comparing abuse history for students who met criteria for each of these variables vs. those who didn’t?

Response 1: We would like to thank the reviewer for this comment. What is meant by additive when combined is simply the expected assault to increase when we add factors. For example, a student-athlete without sleep difficulties should have a lower odds ratio of assault compared to a student-athletes with sleep difficulties. The student-athlete with sleep difficulties should have a lower odds ratio of assault compared to a student-athlete with sleep difficulties that drinks heavily.

We agree with this reviewer that it would be interesting to investigate the factor that may deteriorate sleep.

Comment 2: Statistical Analyses section: There are several inconsistencies regarding with regard to the type of analyses used and description of the interaction terms. First, the Abstract notes that hierarchical logistic regressions were used, which would be appropriate given the clustering of students within schools, while Section 2.3 notes binomial logistic regressions. Second, in Lines 110-111 the authors note that they tested whether “athlete status interacts with sleep difficulties on assault…” and in line 114 they note “an interaction term for each assault variable by student-athlete status.” These statements suggest different things. Further, I am confused why alcohol use and depressive mood are listed as outcome variables. This is not how they are presented in the study aims, nor in the results. It is critical that this paragraph state exactly what how you conducted your analyses, what your predictor and outcome variables are, what variables are being tested as moderators, and what the covariates are for your analyses. At present, this is not clear.

Response 2: We would like to thank the reviewer for these comments. First, we do apologize for the inconsistencies between the abstract and the section 2.3. The reviewer is right when he states that it should have been hierarchical logistic regressions and not binominal. We have deleted “binominal” and replaced it by “hierarchical” in section 2.3.

Comment 3: Results: The section headings for 3.3 and 3.4 are unclear. Did the authors test 3-way interactions with sleep X athlete status X depression (or alcohol)? That is what those titles, and the corresponding table titles indicate. Further, the interaction results are confusing. The authors note significant interactions by depressed mood, alcohol use, and athlete status, which is why they note that they stratified the results, yet Tables 3-5 and corresponding text present data that suggest the relationship between sleep and abuse was significant across depressed/non depressed groups, all alcohol categories, and both athletes and non-athletes. What should we make of this? What information did we learn from the stratified analyses? Did the authors test formal moderation, for example, using the guidelines put forth by Aiken and West in their seminal text on moderation? (Aiken, L. S., & West, S. G. (1991). Multiple regression: Testing and interpreting interactions. Newbury Park, CA: Sage). Additionally, throughout the Results section, I would refer to odds ratios that are greater than 2 by using “2-fold greater odds,” as results such as “102%” are difficult for readers to interpret. Finally, p values in the text are not necessary, as they are stated in the tables.

Response 3: P values have been withdrawn from the text. Every increased likelihood of 100% and higher have been modified to 2-fold greater odds.

Comment 4: The tables must be more clear regarding what is the predictor and what is the outcome and add a note with the covariates and the reference group. It would be helpful to including column headings such as “Abuse Outcomes” and “Sleep Difficulties”. Currently, the abuse variables as presented as if they are the predictors, and it is unclear what the “unadjusted, adjusted, fully-adjusted” columns refer to. Further, the titles are confusing. Tables 2-4 do not show results broken down by “student-athlete status.”

Response 4: We would like to thank the reviewer for this comment. We have adjusted the table and broke it down per student status. We only have modified the title of the tables and hopes they are now clearer for the readerships.

Additional response to the major comments

We would like to thank the reviewer for his comments. We have considered them seriously and decided to review our statistical analyses. We believed the comments were fair and just which led us to enhance the presentation of our results. We hope the following explanations will undo any unnecessary confusion in regard of our results.

First, we have included the rational behind the interactions “To evaluate whether the relationship between sleep and outcomes depended on depressed mood, sleep by depression interactions were evaluated. Significant sleep by depression interactions (p<0.01) were seen for fight, physical assault, threat, touch, sexual assault, and stalking. To evaluate whether the relationship between sleep and outcomes depended on alcohol intake, sleep by alcohol interactions were evaluated. Significant sleep by alcohol interactions (p<0.01) were seen for physical assault, threat, touch, rape, and stalking. Although there were no significant sleep by athlete interactions on outcomes (except for the case of physical assault), we chose to report results stratified by athlete status alongside combined results. The combined results reflect the findings that, in general, relationships did not statistically differ by athlete status. However, given the unbalanced nature of the sample (only about 8% being athletes) and the heuristic value of being able to specifically describe these two groups separately, we have included results stratified by athlete status.”

Given the number of interactions, we have decided not to report them has we believed it would not have enhanced the manuscript. With the stratification by athlete status being clear now in every tables, we hope to have clarify the manuscript. Moreover, with the addition of athlete status in table 2-4, we have deleted table 5 has it was not enhancing the information of the manuscript anymore. 

ABSTRACT

Comment 5: Line 17 – what is meant by “deteriorate the climate”?

Response 5:    We agree with the reviewer that “deteriorate the climate” may be odd and difficult to understand. We have modified it for “unique life experience that university represents”.

INTRODUCTION

Comment 6:   Please specify the difference in prevalence of abuse/assault between athletes vs. non-athletes (e.g., from the AAU 2015 report or the Frintner article, etc.).

Response 6: Thank you for this remark. We agree that this information will enhance the introduction. We have added “and 15% of perpetrators were student-athletes”  

Comment 7:   Lines 41-42: clarify if there is an overrepresentation of student-athletes as perpetrators or victims (or both?).

Response 7:    We would like to thank the reviewer for this comment. We have added “It has been reported that 35% of the perpetrators were student-athletes while they only account for 3% of the entire student population”. With this addition we hope to have clarify the perpetrator/victim question.

Comment 8:   The Introduction describes data on perpetration of assaults, which is interesting but somewhat muddies the waters. I found myself not sure if you were studying victimization, perpetration, or both when I got to your hypotheses. I would consider revising the Introduction to be more clear and perhaps removing data on perpetration if it is not directly related to your hypotheses. This would also clarify what the authors mean by “increased likelihood of abuse” (line 71). Further, lines 75-76 refer to assaults within the past year – is that also the timeframe used in the first hypothesis?

Response 8:    We would like to thank the reviewer for this comment. The data looked at the likelihood of assault from a victim perspective. For example, the question of threat “Were you verbally threatened” Yes or No? Therefore, we do agree that the sentence on Line 74 “increased likelihood of a history” may be misleading. Therefore, we have changed “a history” to “being a victim” in order to make our hypothesis explicit and clear to the readership.

Comment 9:   Lines 69-70, 104: the phrase “poor cognition functioning” is odd, please clarify what this means and/or revise.

Response 9:    Thank you for this comment. We agree with this reviewer that poor cognition functioning is odd. We have changed this for “an increased disinhibition”. Poor sleep, depressive mood and alcohol consumption all have an impact on an individual ability to make a decision. We believe that disinhibition may increase the likelihood of an individual to find himself/herself in an assault situation.

METHODS

Comment 10: Why did you not adjust for race, given known differences in sleep and abuse by racial/ethnic background?

Response 10: Thank you for this comment. We do agree that the addition of ethnic background could have enhance this manuscript. However, the primary objective of this manuscript is to establish the prevalence of these relationships among the student population. We believe that our manuscript should pave the way to investigate the kind of question raised by this reviewer. As reviewer #1 suggested, types of sport will also be an interesting variable to add, ethnicity is definitely a very important variable to be added for future research to further our understanding of these relationships in the student population.

Comment 11: What was the rationale for categorizing alcohol use as stated?

Response 11:    Given the retrospective nature of our data, we used the categories provided by the ACHA-NCHA. The National Institute on Alcohol Abuse and Alcoholism defines binge drinking as a pattern of drinking that brings blood alcohol concentration (BAC) levels to 0.08 g/dL. This typically occurs following 4 drinks for women and 5 drinks for men in about 120 minutes. Therefore, we decided to extend by one drink to be less restrictive since we have no indication on the type of drinks that were consumed (beer, wine, spirits). There was no precise indication on how to separate this variable unless you could test the blood alcohol concentration. We believe that between 7-16 drinks could be considered binge drinking and 17 or more should be considered excessive binge drinking. We have added in our limitation “In addition, consideration should be given to dividing the type of alcohol into three categories (beer wine and spirits). These categories could beneficial for a better understanding of the relationship between student, student-athletes and alcohol consumption”. We believe that this variable could help improve the association and relation between student, student-athletes and alcohol.

DISCUSSION

Comment 12:   8, Lines 41-45: I do not agree with the authors’ statements. Based on your results, the odds ratios for the relationship between sleep and abuse outcomes were almost identical (and all significant) across groups based on depression, alcohol, and athlete status. Thus, it is not accurate to say that those who did not drink or those who were not depressed showed a stronger relationship between sleep and certain categories of abuse. What are we to take away from these findings? They are directly in conflict with the rationale and aims of your study.

Response 12:    We would like to thank the reviewer for this comment. Based on your previous comments addressing our methods and statistic analyses, which we have took very seriously, we have re-run our models and have changed the interpretations as well. We made it clearer that the intent was to compare the student status (student-athlete vs non-athlete). Therefore, a lot of the discussion has changed following careful consideration of your comments.

Comment 13:   Line 97 – the authors refer to “sleep items,” but it appears that a single item was used in analysis. Along these lines, the use of single item should also be mentioned as a study limitation.

Response 13:    Thank you for this comment. We have corrected this oversight. The reviewer is right when he states that sleep was assess with a single item and not several items. We also now made it clear in the limitation that we only used a single sleep item.

Comment 14:   I appreciate the authors’ mention in Line 127 that although the differences between athletes and non-athletes were statistically significant (as would be expected from the large sample size), they were clinically small. Indeed, the raw percentages for most sample characteristics are quite similar for all variables. This needs to be expanded upon as a limitation and the statement in the Discussion (p. 8, Lines 37-40) needs to be tempered accordingly.

Response 14:    Thank you for this comment. We have rephrased the statement in our discussion. Even if we did benefit from a very large simple with anticipated significant results, we believe that the slightest improvement regarding mental health needs to be considered seriously. In that regard, we agree with the reviewer that the collegiate student population would benefit from a thorough investigation on these relationships and that our results should be interpreted as a first step into a translational application for the student population. We hope our results will trigger further research to deepen our understanding of these relationships. 

Round 2

Reviewer 2 Report

IJERPH-1023751 “Verbal, physical, and sexual assault experiences among student-athletes and non-athletes: Impact of sleep, mood, and alcohol use”

I appreciate the authors’ consideration of a number of my comments and I believe they have improved the paper in some areas. However, I do not feel all of my comments were adequately addressed, as noted below.

1. Author responses 2-3 did not adequately address my previous comments. I am still confused why alcohol use and depressive mood are listed as outcome variables. Am I not correct that abuse history is the outcome variable in all analyses, and alcohol and depressive mood were tested as modifiers and then used to stratify analyses? Further, I am still concerned about how the athlete vs. non-athlete analyses and results are described throughout the paper. The authors themselves note in lines 144-154 that there was only 1 statistically significant interaction with athlete status (which is likely due to chance), yet they still chose to present results stratified by athlete status. On the one hand, I can appreciate the author’s aim to present athletes as a separate group, potentially as a way to inspire future research, but I think the fact that athletes are not meaningfully different from non-athletes needs to be noted throughout the paper and language about the strength of results in athletes needs to be notably toned down across the paper.

2. My previous comment #4 was not addressed, with the exception that the authors included table notes with covariates, which I appreciate. The section and table titles still need revision. As they are written, it looks like abuse status is the predictor and sleep is the outcome. If I am understanding correctly, a more accurate and comprehensive section/table title might be, for example (Table 3), “Association between sleep difficulties and history of abuse by depressed mood by athlete status.” Also, please defined “Combined” in the table note or use a more descriptive word, e.g. Full Sample, Combined Sample.

3. Similar to my comment #1 above, I continue to disagree with the authors’ interpretations of some results. First, the authors state throughout the paper that athletes look worse (for one example, lines 86-106). The OR for all analyses were similar, which supports the idea that these associations are important for college students as a whole, not only for athletes. The authors did not adequately address this in the paper or discussion. I also disagree that student athletes who do not report depression report more abuse – according to the tables, these relationships are significant in both depressed and non-depressed groups, and in both athletes and non-athletes alike.

Additionally, in Line 122-123 the authors note: “As previously mentioned, the number of beverages impacted inversely the likelihood of being involved in a fight for student-athlete and non-athlete.” I do not think the authors can make anything of the fact that the odds increased slightly across increasing alcohol categories for non-athletes, while the odds decreased slightly across alcohol categories for non-athletes, if I am interpreting their statement correctly. The results were significant in both groups, and these stratified analyses do not tell us anything about whether the differences between athletes/non-athletes or across alcohol categories are statistically significant or meaningful, only that there are associations between poor sleep and abuse history within all groups/categories that were studied (which is confusing in its own right).  

4. While the authors answered my comment 11, it would be important to add some of that information to the text, e.g., the citation for where they got the categories.

Minor comments:

5. Measures – does non-athlete status include those who participated in club sports and intramurals, as well as those who participated in no sports? Please clarify in text.

Author Response

IJERPH-1023751 “Verbal, physical, and sexual assault experiences among student-athletes and non-athletes: Impact of sleep, mood, and alcohol use”

I appreciate the authors’ consideration of a number of my comments and I believe they have improved the paper in some areas. However, I do not feel all of my comments were adequately addressed, as noted below.

We would like to thank the reviewer for this comment and also taking the time to do a second revision. We have, in order to enhance the manuscript, changed the title of the manuscript which we believed add clarity for the readership.

  1. Author responses 2-3 did not adequately address my previous comments. I am still confused why alcohol use and depressive mood are listed as outcome variables. Am I not correct that abuse history is the outcome variable in all analyses, and alcohol and depressive mood were tested as modifiers and then used to stratify analyses? Further, I am still concerned about how the athlete vs. non-athlete analyses and results are described throughout the paper. The authors themselves note in lines 144-154 that there was only 1 statistically significant interaction with athlete status (which is likely due to chance), yet they still chose to present results stratified by athlete status. On the one hand, I can appreciate the author’s aim to present athletes as a separate group, potentially as a way to inspire future research, but I think the fact that athletes are not meaningfully different from non-athletes needs to be noted throughout the paper and language about the strength of results in athletes needs to be notably toned down across the paper.

We would like to thank the reviewer for this comment. We do agree with the reviewer that the tone which was used to describe and interpret results may have been misleading. Therefore, we have reviewed the discussion and conclusion sections accordingly. As the reviewer pointed out, student-athletes represent a subpopulation of students that needs to be investigated separately. Throughout the manuscript we have made an attempt to focus equally on student non-athletes and student-athletes since there was only 1 statistical interaction as pointed out by the reviewer. We believed the changes will nuance the results and will satisfy the reviewer.  Again, we believe that this comment from the reviewer has helped the authors to clarify the content and the results of this manuscript. In addition, in the statistical analyses section we also have clarified the variables as followed; 1) sleep is the independent variable; 2) assaults are the outcomes and the modifiers are the alcohol intake and depressed mood.

  1. My previous comment #4 was not addressed, with the exception that the authors included table notes with covariates, which I appreciate. The section and table titles still need revision. As they are written, it looks like abuse status is the predictor and sleep is the outcome. If I am understanding correctly, a more accurate and comprehensive section/table title might be, for example (Table 3), “Association between sleep difficulties and history of abuse by depressed mood by athlete status.” Also, please defined “Combined” in the table note or use a more descriptive word, e.g. Full Sample, Combined Sample.

We would like to thank the reviewer for this comment and apologize for the lack of clarification in our previous revision. We have changed the way we report our results as previously mentioned. Therefore, we have changed the titles of every tables except table 1. In addition, with the modification in statistical analyses, we believe it has now clarified the point of the reviewer. The outcome are the assaults and sleep is the independent variable. We have added in the title when the results were controlled for depressed mood and alcohol use. We believe these changes will enhance the presentation of our results. In addition, we have clarified “combine”, which should now clearly represent the full sample of students (non-athletes and student-athletes).

  1. Similar to my comment #1 above, I continue to disagree with the authors’ interpretations of some results. First, the authors state throughout the paper that athletes look worse (for one example, lines 86-106). The OR for all analyses were similar, which supports the idea that these associations are important for college students as a whole, not only for athletes. The authors did not adequately address this in the paper or discussion. I also disagree that student athletes who do not report depression report more abuse – according to the tables, these relationships are significant in both depressed and non-depressed groups, and in both athletes and non-athletes alike.

We would like to thank the reviewer for this comment. We do agree that the OR is similar for student-athletes and non-athletes. To this point, we have decided to focus on both the student non-athletes and student-athletes and have modified considerably our approach. We still maintain that even if the differences between student-athletes and student non-athletes are not statistically significant it does not mean it is not clinically significant. We believe that when it comes to mental health, the slightest change should be reported and taken seriously has the literature remains sparse in the collegiate population. However, we do agree with the reviewer that we should have acknowledged the differences between those with and without perceived sleep for both the student-athletes and student non-athletes. This has now been corrected. We have changed our approach for the alcohol use and depressed mood by clearly stating the absence of statistical differences. We hope that the change of approach and a more nuanced approach will satisfying to the reviewer.

Additionally, in Line 122-123 the authors note: “As previously mentioned, the number of beverages impacted inversely the likelihood of being involved in a fight for student-athlete and non-athlete.” I do not think the authors can make anything of the fact that the odds increased slightly across increasing alcohol categories for non-athletes, while the odds decreased slightly across alcohol categories for non-athletes, if I am interpreting their statement correctly. The results were significant in both groups, and these stratified analyses do not tell us anything about whether the differences between athletes/non-athletes or across alcohol categories are statistically significant or meaningful, only that there are associations between poor sleep and abuse history within all groups/categories that were studied (which is confusing in its own right). 

Again, we would like to thank the reviewer for this thoughtful comment. We now mention that even with a linear increase “that none of these categories have reach a statistical difference between student-athletes and student non-athletes.” We feel it is important to mention this trend or increased OR event if it does not differ significantly statistically between student non-athlete and student-athletes. Still, we do agree with the reviewer that we should have clarified that this was not statistically different. Throughout the discussion, we now equally discuss the non-athlete and student-athletes results and put the emphasis on the quasi absence of differences.

  1. While the authors answered my comment 11, it would be important to add some of that information to the text, e.g., the citation for where they got the categories.

We would like to thank the reviewer for this comment. We agree that the definition of binge drinking should have been mentioned. It has therefore been added in the introduction (reference #16).

Minor comments:

  1. Measures – does non-athlete status include those who participated in club sports and intramurals, as well as those who participated in no sports? Please clarify in text.

We would like to thank the reviewer once again for highlighting an oversight from our part.   We have added in the Measures section that only those who selected the option “Varsity” were considered student-athletes.